# Comparing regular expression and machine learning approaches to predict immigrant status from primary care electronic medical record data in Ontario, Canada

Stephanie Garies[1], Christopher Meaney[2], Karen Weyman[2,3], Gary Bloch[2,3], Jessica Gronsbell[2,4,5], Nassim Vahidi-Williams[3], Ri Wang[6], Noah Crampton[2,7], Karen Tu[2,7,8], Andrew D. Pinto[2,3,6,9]*

1 Department of Family Medicine, University of Calgary, Calgary, Alberta, Canada, 2 Department of Family & Community Medicine, Faculty of Medicine, University of Toronto, Toronto, Ontario, Canada, 3 Department of Family & Community Medicine, St. Michael's Hospital, Toronto, Ontario, Canada, 4 Department of Statistical Sciences, University of Toronto, Toronto, Ontario, Canada, 5 Department of Computer Science, University of Toronto, Toronto, Ontario, Canada, 6 Upstream Lab, MAP Centre for Urban Health Solutions, Li Ka Shing Knowledge Institute, St. Michael's Hospital, Toronto, Ontario, Canada, 7 Toronto Western Family Health Team, University Health Network, Toronto, Ontario, Canada, 8 North York General Hospital, Toronto, Ontario, Canada, 9 Dalla Lana School of Public Health, University of Toronto, Toronto, Ontario, Canada

* andrew.pinto@utoronto.ca

## Abstract

New immigrants often face barriers when navigating the healthcare system, which can create unmet healthcare needs and contribute to health inequities. Primary care practices, as the gateway to the healthcare system, could use information about their patients' immigrant status to ensure accessible care and equitable resource allocation. However, this is not routinely collected or documented in primary care. The objective of this study was to explore two approaches (regular expression and machine learning) to determine patient-reported immigrant status from primary care electronic medical records (EMRs). De-identified EMR data from the St. Michael's Hospital Academic Family Health Team in Toronto, Ontario, Canada was used, including the reference set of patient-reported responses to a health equity question-naire. Two approaches were tested and compared: 1) a regular expression classifier (using key text terms), and 2) supervised machine learning classifier (specifically XGBoost). Discrimination and calibration metrics were calculated using self-reported immigrant status from the patient surveys. Among eligible patients in the analytic cohort (N = 12,998), 44.5% reported being born outside of Canada. Although the XGBoost model outperformed the regular expression approach (XGBoost sensitivity = 53.1% and positive predictive value = 72.6%; regular expression sensitivity = 5.2% and positive predictive value = 96.8%), neither approach was accurate enough for use in practice. While understanding patients' immigrant status is important for the provision of high quality, comprehensive primary health care, our work demonstrates

**Data availability statement:** The EMR data used in this study are only available for approved research projects through Unity Health Toronto (upstreamlab@unityhealth.to or https://upstreamlab.org/).

**Funding:** This study was funded through the Canadian Institutes of Health Research (CIHR) Health System Impact Fellowship program, in collaboration with Unity Health Toronto (SG).

**Competing interests:** The authors have declared that no competing interests exist.

the challenges of using EMR data to derive immigrant status. For now, primary care practices should continue to rely on obtaining immigrant status through initial patient intakes or surveys.

---

## Author summary

Healthcare providers often lack important information about social determinants of health for their patients. For instance, understanding immigrant status for patients could help tailor more comprehensive care, as well as determine equitable resource support and allocation. Using artificial intelligence (AI) to predict immigrant status is a novel approach that could help providers and healthcare systems understand needs in their community. We tested a machine learning approach and a simpler regular expression (key term) search to investigate how well we could identify immigrant status using data from primary care electronic medical records. The machine learning model performed slightly better than the key term search, however, neither were adequately accurate for use in clinical practice. While more advanced AI approaches could be tested, we caution against introducing bias particularly for a sensitive attribute such as immigrant status and recommend continued capture of this information directly from patients.

## Introduction

In Canada, primary care is often the first point of entry into the healthcare system. Although Canada has a universal, publicly funded healthcare system, access to primary care is not always equitable. New immigrants in Canada, in particular, face a wide range of barriers to care, such as language and cultural differences, and lack of information about accessing or navigating the healthcare system [1]. Subsequently, this can create unmet health care needs for immigrant populations, as well as contribute to health inequities [1–3].

Understanding patient immigrant status could assist primary care practices with ensuring equitable and easily accessible care. Unfortunately, this information is not captured routinely or consistently in the electronic medical record (EMR) systems used in primary care settings. Asking patients directly about social determinants of health (SDoH), including immigrant status, has been shown to produce good response rates [2]. However, this can be time-consuming and resource intensive for busy clinical practices, and it is unlikely that this information will be captured for all patients in a clinic or recorded consistently in the EMR.

A novel and less intrusive approach could make use of the rich data within EMRs to identify or derive patients' immigrant status, which could then be used for more tailored patient care, practice management and program planning. Artificial intelligence (AI), such as machine learning, is a new approach that has not been widely tested in EMR

data for this purpose, with a few examples using health surveys from the United States [4] and other non-clinical databases. Despite the promise of AI, using EMR data to derive immigrant status is laden with important issues related to ethics, privacy, consent and bias [5–7], and necessitates careful consideration of the development and use of these algorithms.

The objective of this study was to explore different approaches (i.e., regular expression and machine learning) for determining information about immigrant status using self-reported responses from a patient survey.

## Methods

### Setting & study population

Unity Health Toronto is a large academic health science centre in downtown Toronto, Ontario, Canada with an emphasis on serving individuals and communities made vulnerable by social and economic policies. It is home to the St. Michael's Hospital Academic Family Health Team (SMHAFHT), consisting of approximately 80 family physicians and teams of allied health professionals, such as nurses, social workers, pharmacists, and income support facilitators, who work across multiple sites in downtown east Toronto.

### Data source

The SMHAFHT EMR system captures demographic, social, and clinical information for patients in the practice. Previous work by the region's Local Health Integration Network, and a more recent project led by Pinto et al., [2] focused on collecting SDoH information using an 11-item electronic survey ('Health Equity Questionnaire' [HEQ]) completed by patients in the waiting rooms of the SMHAFHT. The HEQ includes questions about preferred language, immigrant status, racial or ethnic group, disabilities, gender identity, sexual orientation, income, religion, and housing status. Since 2013, approximately one third of patients (~16,500) have had their SDoH information collected and integrated into their EMR as a free-text progress note. The remaining two thirds of patients may have some SDoH information in various fields in their record, but much of this is likely to be found in unstructured narrative/clinical notes with variable completeness.

The EMR data extracted for this study contained structured or semi-structured data (patient demographics, physician billing, visits, etc.) and unstructured progress notes made by the primary care team. Data were de-identified prior to analysis using Python software (Python Software Foundation, http://www.python.org). The de-identification process employed a pattern-matching approach to remove potential identifiers (e.g., names, phone numbers, etc.) that may have been present in other text fields in the EMR data.

### Reference standard

The HEQ asks patients if they were born in Canada with options for a *Yes* or *No* response. Responses indicating *Don't Know* or *Prefer Not to Answer* were removed (N = 164; S1 Fig). The most recent response to this question was used as a binary indicator of immigrant status, which constituted our outcome measure.

### Cohort & descriptive analysis

A descriptive summary of patient demographic and clinical characteristics was reported for all eligible patients who had at least 1 clinic visit between January 1, 2013 (start of the HEQ data collection) and November 15, 2021 (date of the research ethics submission), and who were not deceased or had transferred to another practice. This included sex (as reported in the patient demographic table in the EMR, restricted to values of male or female), and mean age in years. To be included in the analytic cohort, patients were selected if they had a valid *yes*/*no* response to the immigrant question in the HEQ. Patients were also included if they had at least one primary care diagnostic code, primary care service code (indicating a visit took place) and primary care clinical note in their EMR data. Only EMR data that occurred up to and including the date of the patient's most recent HEQ response was used in the analysis.

## Feature engineering

Three high-dimensional features were used from the primary care EMR dataset: family medicine diagnostic codes (from the Ontario Health Insurance Plan [OHIP] [8], which is a derivation of the International Classification of Diseases version 9 [ICD-9]), service codes (also from OHIP [9]), and clinical text data from progress notes. For each variable type (diagnostic codes, service codes, and clinical text data), a vector space modelling approach [10] was used to engineer sparse features. Code-frequency/term-frequency vectors were constructed at a patient-level. The dimension of the diagnostic-code-frequency-vector is equal to the number of unique diagnostic codes observed over all patients in the billing table. Similarly, the dimension of the service-code-frequency vector is equal to the number of unique service codes observed over all patients in the billing table. Lastly, the dimension of the term-frequency vector is equal to the number of unique words/tokens observed over the entire clinical note corpus. Code-frequency/term-frequency vectors were concatenated into a matrix (with the row dimension representing unique patients, and the column dimension representing diagnostic codes, services codes or words/tokens, respectively). The resulting patient-code matrices and patient-term matrices are high-dimensional sparse matrices. Each individual element of the matrix is a count of the number of times 1) a diagnostic code, 2) a service code, or 3) a word/token was observed in the patient EMR (billing data or clinical notes). A total of 124,574 unigram words/tokens, 438 diagnostic codes, 272 service codes and 2 demographic features (patient age and sex) were included.

## Statistical model

The first approach aimed to train and evaluate a supervised machine learning model for prediction of a binary outcome (immigrant status) as a function of patient diagnostic codes, patient service codes, and clinical text data. Several machine learning models were considered (e.g., logistic regression and its regularized variants, feed-forward neural networks, random forests, and extreme gradient boosting models). Ultimately, XGBoost models were determined to be most performant for this particular prediction task and was used for the analysis reported in this paper. Python XGBoost implementation of the extreme gradient boosting classifier was used [11]. This is an ensemble model that uses sequentially weighted tree-based sub-models and a Bernoulli/cross-entropy loss function optimized over a specified number of fitting rounds. Each sub-model in the sequential learning process is weighted by a learning rate. Each specific sub-model is a decision tree, optimizing Bernoulli/cross-entropy loss. Maximum tree depth and minimum leaf weight control the complexity/depth of each sequentially learned tree. Gamma, alpha and lambda regularization are used to further control complexity of the learned trees. Column sampling is used to select subsets of features for inclusion in the tree building process, whereas row subsampling is used to select subsets of patients for inclusion in the tree building process. Details regarding each hyper-parameter are provided in the table below, as well as the API documentation. The "total coverage" metric was used to assess variable importance.

## Hyper-parameter optimization methods

A tree-Parzen estimator hyper-parameter optimization strategy (implemented using the Python *hyperopt* package [12]) was used to optimize model performance. The hyper-optimization experiment was designed with a computational budget of N = 50 trials. For each individual trial in the overarching experiment (n = 1…50), an extreme gradient boosting model was fit at a unique hyper-parameter configuration on the randomly selected training dataset (N = 7,808), optimizing a Bernoulli/cross-entropy loss function. The performance of each of the N = 50 trained models was evaluated on a randomly selected validation dataset using an area under the receiver operating characteristic curve (AUC) performance metric. S1 Table provides more details on the specific hyper-parameter configurations.

## Evaluation metrics and design

The training, validation and test data were randomly split as 60:20:20, respectively. Extreme gradient boosting models were trained on the training dataset, with model performance estimated using the independent (randomly sampled)

validation dataset. Generalization performance was evaluated on the held-out test dataset (internal validation). Several metrics of discrimination and calibration were reported, including sensitivity, specificity, positive predictive value (PPV), negative predictive value (NPV), AUC, receiver operating characteristic (ROC) curve and a LOESS smoothed calibration curve. Model calibration performance was estimated using the integrated calibration index (ICI) and the E50, E90 and EMax indices [13]. Discrimination and calibration statistics were estimated collectively on the overall test set, but also separately for individual age/sex strata in the test set to assess whether model predictive ability was similar across age/sex groups (i.e., whether the model satisfies specific notions of predictive fairness).

### Regular expression and pattern matching methods

A panel of experts in primary care and data science (family physicians, epidemiologists, health services researchers, biostatistician), as well as patient advisors, identified key terms, codes and concepts that were used in the construction of a set of 17 patterns that were applied over the EMR clinical text data. Other sources were also explored to ensure all possible terms and codes were included, such as the Unified Medical Language System [14], diagnostic and service billing codes from the Ontario Ministry of Health [8,9], previous literature and by data exploration (especially for identifying misspellings or abbreviations). Each of the 17 unique regular expressions returned a Boolean flag, which was then combined into a single REGEX classifier (via an *OR* operator). The performance of the REGEX classifiers was evaluated using several discrimination metrics (sensitivity, specificity, PPV and NPV) using the test dataset.

### Software

A combination of Python version 3.10 and R version 3.6 were used for the analysis.

### Ethics statement

This study was reviewed and approved by the Unity Health Toronto Research Ethics Board (REB 21–301). A waiver of individual patient consent was granted for the use of de-identified data and in accordance with the Tri-Council Policy Statement 2 (TCPS2).

## Results

In total, there were 37,171 patients who had at least one visit to the SMHAFHT in the study period (2013–2021) and who were not deceased or inactive. Of these, 16,156 had a valid response to the immigrant question in the HEQ. After excluding those without data prior to the HEQ, 12,998 patients were included in the analysis (Table 1, S1 Fig). These patients tended to be younger and more likely female, as compared to all patients in the clinic (Table 1). Among those who provided an eligible response to the immigrant question, 7,208 (55.5%) individuals indicated that they were born in Canada and 5,790 (44.5%) individuals reported being born outside of Canada.

### Machine learning approach

Table 1 also summarizes the patients included within the machine learning training, validation and test datasets by outcome (immigrant status), age and sex. In the validation dataset, model performance varied considerably as a result of hyperparameter configurations, with the optimal AUC estimated to be 0.731 (S2 Fig; S2 Table). When using the final test dataset, the model suffered from low sensitivity (53.1%, 95% CI 50.2-56.0) and poor calibration (ICI = 0.039) (Table 2, Fig 1).

Features deemed most important in the model were explored briefly but varied with each model run; examples of important features that occurred more frequently included countries (i.e., 'immigrated', 'canada' 'bangladesh', 'phillipines', etc.), patient age, and certain health attributes or conditions (i.e., 'smoking', 'pylori', 'latent', diagnosis code 250 [diabetes]).

**Table 1. Characteristics of all patients and patients included in the analysis cohort.**

| Characteristic | All patients with at least one visit since 2013 N = 37,171 | Patients included in the analysis cohort | | | |
| --- | --- | --- | --- | --- | --- |
| | | Full analytic cohort (N = 12,998) | Training dataset (N = 7808) | Validation dataset (N = 2635) | Test dataset (N = 2555) |
| Sex, n (%) | | | | | |
| Female | 19,810 (53.3) | 7,467 (57.4) | 4,434 (56.8) | 1,529 (58.0) | 1,504 (58.9) |
| Male | 17,361 (46.7) | 5,531 (42.6) | 3,374 (43.2) | 1,106 (42.0) | 1,051 (41.1) |
| Age, mean years (SD) | 50.2 (16.1) | 44.8 (15.1) | 44.9 (15.2) | 44.6 (15.0) | 44.5 (15.1) |
| Born in Canada, n (%) | | | | | |
| No | *n/a* | 5,790 (44.5) | 3,472 (44.5) | 1,154 (43.8) | 1,164 (45.6) |
| Yes | *n/a* | 7,208 (55.5) | 4,336 (55.5) | 1,481 (56.2) | 1,391 (54.4) |

**Table 2. Discrimination and calibration metrics for XGBoost and Regular Expression approaches using the test dataset.**

| | Regular Expression (test dataset) | XGBoost (test dataset) |
| --- | --- | --- |
| **Discrimination Metrics** | | |
| Sensitivity, % (95% CI) | 5.2 (4.0, 6.5) | 53.1 (50.2, 56.0) |
| Specificity, % (95% CI) | 99.9 (99.7, 100) | 83.2 (81.3, 85.2) |
| Positive predictive value, % (95% CI) | 96.8 (92.5, 100) | 72.6 (69.6, 75.6) |
| Negative predictive value, % (95% CI) | 55.7 (53.8, 57.7) | 68.0 (65.7, 70.2) |
| AUC | -- | 74.6 (72.7, 76.5) |
| F1 Score (95% CI) | 9.9 (7.5, 12.4) | 61.3 (58.0, 64.7) |
| Balanced Accuracy (95% CI) | 52.5 (51.9, 53.2) | 68.2 (66.4, 69.9) |
| **Calibration Metrics** | | |
| ICI | -- | 0.039 |
| E50 | -- | 0.031 |
| E90 | -- | 0.115 |
| EMax | -- | 0.136 |

AUC= area under the receiver operating characteristic curve, CI=confidence interval, ICI=integrated calibration index, REGEX=regular expression.

## Regular expression approach

The REGEX classifier using key terms and codes performed worse than the XGBoost model at predicting immigration status (Table 2). The REGEX model rarely classified any patient as being born outside of Canada; therefore, sensitivity was very low (5.2%, 95% CI 4.0-6.5). However, the classifier was often correct in the few patients that were positively identified as being born outside of Canada and therefore PPV (96.8%, 95% CI: 92.5-100.0) and specificity (99.9%, 95% CI: 99.7-100.0) were high.

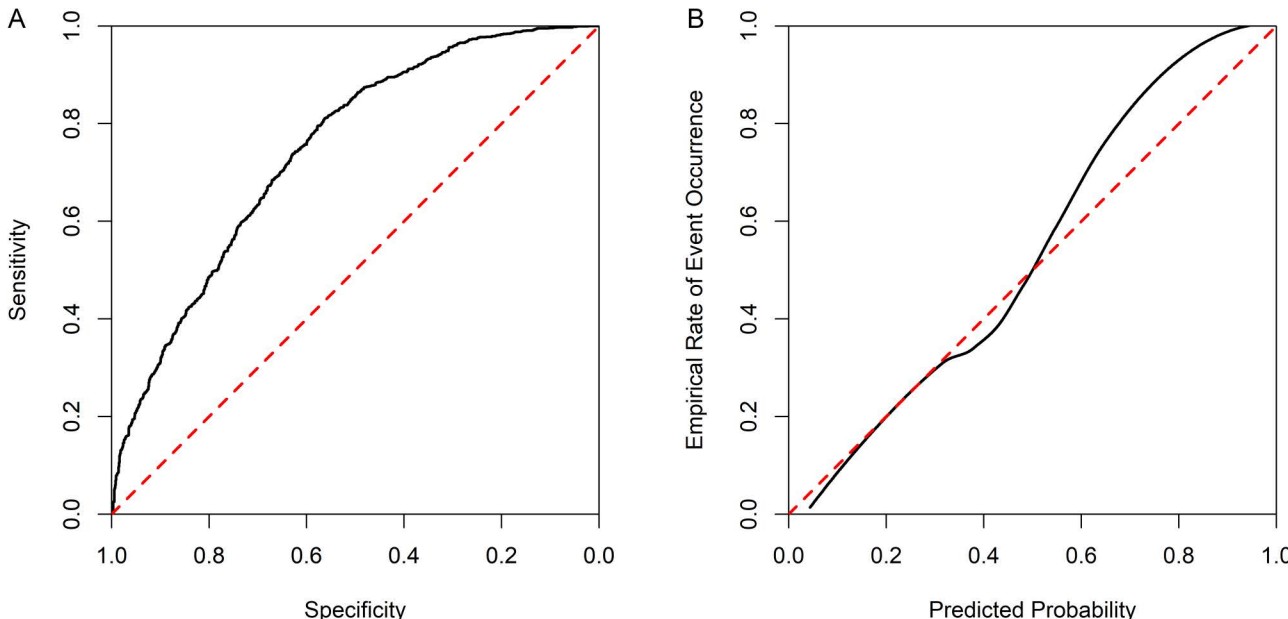

**Fig 1. Receiver operating characteristic (ROC) curve (a), and LOESS smoothed calibration curve (b), estimated on test dataset.**

### Predictive fairness

When investigating demographic differences in accuracy, both the REGEX classifier and machine learning model demonstrated minor variations in performance across sex and age groups; however, in general, the overall patterns of inferences still held within each age/sex strata (i.e., all models had poor sensitivity) (S3 Table, S4 Table).

A brief review of a small random subset of records indicated that false negatives were likely arising from patients with the outcome (i.e., positive immigrant status) who were not being identified by the models due to an absence of data indicating this in the EMR. Conversely, false positives appeared to arise due to either measurement error in capturing immigrant status or semantic ambiguity around the terms that are predictive of immigrant status or when a patient was born/ arrived in Canada.

### Interpretation

This study described two approaches for identifying immigrant patients within primary care EMR data. This exploratory work was driven by the recognition that collecting SDoH information placed a heavy administrative burden on the practice staff and thus, the survey had not been offered consistently to each and every patient, resulting in only one-third of all patients in the practice having completed the HEQ. This may have also impacted the development of our classification model and likely contributed to selection bias in terms of patients who were included in the analysis. However, it appears that both primary data collection and secondary EMR data analysis remain inadequate to capture immigrant status accurately and consistently from patients in primary care settings. The poor overall sensitivity across both the regular expression and machine learning methods highlights this challenge, although this is not an issue exclusive to the SMHAFHT. One of the largest primary care EMR databases in the world (Clinical Practice Research Datalink) found codes representing immigrant status in only 1.6% of nearly 600,000 patients across the United Kingdom [6]. Our regular expression

approach performed poorly in terms of sensitivity, which may be attributed to a lack of documentation or specific text terms referring to immigrant status in the EMR; sensitivity improved when using a machine learning approach, possibly identifying new associations or text terms that had not been considered in the rule-based search (i.e., names of countries).

There are important reasons why patients' immigrant status should be available for clinical use. For example, this would allow clinicians the opportunity to provide culturally safe and appropriate care for medical, social and economic needs and to ensure continuity in team-based care settings, as well as for understanding practice- and population-level characteristics to ensure adequate resourcing, to monitor and address health disparities, and to identify patient groups who may be facing barriers to care [2,3,15]. While the regular expression approach had poor sensitivity, its high PPV (97%) may be useful for identifying a smaller cohort of immigrant patients; however, this would not capture most (due to low sensitivity) and may only be indexing a non-representative subgroup of immigrant patients. A regular expression approach tends to be more easily interpretable for use in clinical practice, which can help build trust and facilitate implementation, but the overall validity for this particular algorithm is inadequate and we do not recommend use in its current form.

Despite the potential clinical and system level benefits, there are also substantial ethical concerns to acknowledge when attempting to identify immigrant status within EMRs, including the potential for discrimination, contact with immigration officials, increased vulnerability of experiencing systemic violence, infringement of privacy and confidentiality, and exacerbating biases perpetuated by the use of machine learning and EMR data [5,16–18]. Furthermore, a focus on immigrant status alone, without additional context around personal, differentiated patient experiences, may lead to essentialism and could hinder interventions or decisions aimed at reducing health inequities that often require a comprehensive, holistic approach [19]. Lastly, there have even been suggestions that the "explicit documentation of immigration status of patients and their family members in a health record should be avoided, particularly when risks outweigh benefits and risks are rapidly changing" [7] (i.e., in reference to policies in the United States). While Canada may differ from the United States in terms of immigration laws, it is still necessary to carefully consider all risks and benefits to patients and clinical teams, including ensuring adequate communication to patients about the purpose of collecting these data and offering the option to decline sharing this information.

## Limitations

There are several limitations of this study. The first is the use of EMR data to derive immigrant status – this is unlikely to be recorded consistently or at all in a busy clinical environment. Further, there may be enough heterogeneity across various immigrant groups that the machine learning model is unable to detect consistent patterns in the EMR data. There are many sophisticated AI approaches now available that were not explore in this study, such as word embedding models, transformers, large language models, enhanced feature engineering, among others. Future work would investigate whether more flexible models, or models with deep capacity/architectures, might be used to improve model performance. Despite the promise of improve accuracy, many of these methods, and large language models in particular, have demonstrated bias and the potential for patient harm [20,21], and should be used cautiously, if at all, in the context of social data. Further, it is important not to distract from efforts to capture these data in a consensual, patient-focused way; for instance, an updated version of the HEQ was recently launched in 2023 within the SMHAFHT and includes the support of a 'Community Surveyor', who will focus on enhancing response rates using an equity-informed approach to data collection. Further work is also required to examine the potential ethical issues and privacy protection when deriving immigrant status from EMR data, and whether resources should instead be redirected to supporting better primary data collection from patients themselves.

Secondly, the reference standard responses from the HEQ may also have introduced selection bias in terms of patients who were willing and able to complete the survey and felt comfortable answering the immigrant question. While this study only compared a limited set of patient characteristics for HEQ respondents and non-respondents, the groups appeared to be somewhat similar (Table 1), although there was a higher proportion of females and younger patients in the analytic cohort compared to all patients with a visit during the study period.

This analysis focused on one large primary care practice using one EMR system in Toronto, Ontario and these analytic strategies may not be easily transferred to data from other EMR systems or jurisdictions. Lastly, this classification analysis treated immigrant status as a binary outcome, yet there are other important aspects that were not considered for inclusion (e.g., time since immigration, country, etc.) and could have more meaningful impact on health outcomes and healthcare access (i.e., newer immigrants are likely to face additional barriers to healthcare [1]), but these approaches were not designed to capture this information. The proportion of patients with and without the outcome was not significantly imbalanced (i.e., 44.5% not born in Canada vs. 55.5% born in Canada), however we did not apply techniques to address possible issues with imbalanced cases (e.g., SMOTE, over/under-sampling, class weighting, etc.) and this could be tested in future iterations.

## Conclusion

In examining two methods for identifying immigrant status in primary care EMR data, both the regular expression and machine learning approaches were limited in their usability in clinical practice or for practice- or population-level applications. Given this and the concerns around ethics, patient privacy and bias, it may not be feasible and/or appropriate to derive immigrant status using these approaches on clinical data, but rather encourage the generation of this information from patients directly through surveys or during the initial intake visit.

## Supporting information

**S1 Fig. Patient flow diagram.**
(TIFF)

**S2 Fig. AUC estimates for each of the N = 50 budgeted hyper-parameter optimization experiments using the validation dataset.**
(TIFF)

**S1 Table. Hyper-parameters associated with the extreme gradient boosting (XGBoost) model, and the generating distributions used to sample hyper-parameter values for each hyper-parameter optimization experiment.**
(DOCX)

**S2 Table. Validation dataset AUC metric, hyper-parameter optimization time, and optimally identified hyper-parameter configurations for the simulated annealing experiments, applied to the binary XGBoost classifier.**
(DOCX)

**S3 Table. Discrimination statistics for the XGBoost model stratified by age group and sex.**
(DOCX)

**S4 Table. Discrimination statistics for the REGEX/pattern-matching classifier stratified by age group and sex.**
(DOCX)

## Author contributions

**Conceptualization:** Stephanie Garies, Karen Weyman, Gary Bloch, Jessica Gronsbell, Ri Wang, Noah Crampton, Karen Tu, Andrew D. Pinto.

**Data curation:** Stephanie Garies, Ri Wang, Andrew D. Pinto.

**Formal analysis:** Stephanie Garies, Christopher Meaney.

**Funding acquisition:** Stephanie Garies, Karen Weyman, Andrew D. Pinto.

**Investigation:** Stephanie Garies, Christopher Meaney, Jessica Gronsbell, Ri Wang.

**Methodology:** Stephanie Garies, Christopher Meaney, Karen Weyman, Gary Bloch, Jessica Gronsbell, Nassim Vahidi-Williams, Ri Wang, Noah Crampton, Karen Tu, Andrew D. Pinto.

**Project administration:** Stephanie Garies, Andrew D. Pinto.

**Resources:** Karen Weyman, Andrew D. Pinto.

**Software:** Stephanie Garies, Christopher Meaney, Ri Wang.

**Supervision:** Stephanie Garies, Karen Weyman, Gary Bloch, Jessica Gronsbell, Nassim Vahidi-Williams, Noah Crampton, Karen Tu, Andrew D. Pinto.

**Validation:** Stephanie Garies, Christopher Meaney.

**Writing – original draft:** Stephanie Garies, Christopher Meaney.

**Writing – review & editing:** Stephanie Garies, Christopher Meaney, Karen Weyman, Gary Bloch, Jessica Gronsbell, Nassim Vahidi-Williams, Ri Wang, Noah Crampton, Karen Tu, Andrew D. Pinto.

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
