## [Decision Letter · Decision Letter 0]

4 Mar 2025

Response to Reviewers'. This file does not need to include responses to any formatting updates and technical items listed in the 'Journal Requirements' section below.'. This file does not need to include responses to any formatting updates and technical items listed in the 'Journal Requirements' section below.* A marked-up copy of your manuscript that highlights changes made to the original version. You should upload this as a separate file labeled 'Revised Manuscript with Track Changes'.'.* An unmarked version of your revised paper without tracked changes. You should upload this as a separate file labeled 'Manuscript'.'. If you would like to make changes to your financial disclosure, competing interests statement, or data availability statement, please make these updates within the submission form at the time of resubmission. Guidelines for resubmitting your figure files are available below the reviewer comments at the end of this letter. We look forward to receiving your revised manuscript. Kind regards, Gaurav Laroia, Ph.D.Section EditorPLOS Digital Health Samira Abbasgholizadeh RahimiAcademic EditorPLOS Digital Health Leo Anthony CeliEditor-in-ChiefPLOS Digital Healthorcid.org/0000-0001-6712-6626 **Journal Requirements:**1. Please provide an Author Summary. This should appear in your manuscript between the Abstract (if applicable) and the Introduction, and should be 150–200 words long. The aim should be to make your findings accessible to a wide audience that includes both scientists and non-scientists. Sample summaries can be found on our website under Submission Guidelines:

https://journals.plos.org/digitalhealth/s/submission-guidelines#loc-parts-of-a-submission

**Additional Editor Comments (if provided):****Reviewers' Comments:** Reviewer's Responses to Questions

**Comments to the Author**

1. Does this manuscript meet PLOS Digital Health’s publication criteria? Is the manuscript technically sound, and do the data support the conclusions? The manuscript must describe methodologically and ethically rigorous research with conclusions that are appropriately drawn based on the data presented.? Is the manuscript technically sound, and do the data support the conclusions? The manuscript must describe methodologically and ethically rigorous research with conclusions that are appropriately drawn based on the data presented.

Reviewer #1: Yes

Reviewer #2: Yes

2. Has the statistical analysis been performed appropriately and rigorously?

Reviewer #1: Yes

Reviewer #2: Yes

3. Have the authors made all data underlying the findings in their manuscript fully available (please refer to the Data Availability Statement at the start of the manuscript PDF file)?

The PLOS Data policy requires authors to make all data underlying the findings described in their manuscript fully available without restriction, with rare exception. The data should be provided as part of the manuscript or its supporting information, or deposited to a public repository. For example, in addition to summary statistics, the data points behind means, medians and variance measures should be available. If there are restrictions on publicly sharing data—e.g. participant privacy or use of data from a third party—those must be specified.requires authors to make all data underlying the findings described in their manuscript fully available without restriction, with rare exception. The data should be provided as part of the manuscript or its supporting information, or deposited to a public repository. For example, in addition to summary statistics, the data points behind means, medians and variance measures should be available. If there are restrictions on publicly sharing data—e.g. participant privacy or use of data from a third party—those must be specified.

Reviewer #1: Yes

Reviewer #2: Yes

4. Is the manuscript presented in an intelligible fashion and written in standard English?

Reviewer #1: Yes

Reviewer #2: Yes

Reviewer #1: The study could explore some data balancing techniques (SMOTE/class weighting) and label validation. Adding more discussions for the FP/FN cases could help to identify patterns in ambiguity, missing context/data, or label noise. Other metadata, such as time since immigration, country, etc., definitely could have a big impact on the model's performance; I would suggest including accuracy and F1 score to have a more holistic evaluation.

Please correct “e., time since immigration” in Line 366, to "e.g. time since immigration"

Reviewer #2: 1. Selection bias in training data constitutes major limitation of this study. Researchers included patients who completed Health Equity Questionnaire (HEQ), representing portion of total patient population (16,500 out of 37,171 patients). Selective sampling raises concerns about representation, generalizability. Study authors acknowledge individuals who completed surveys might differ from those who didn't. Study population was younger with higher proportion of females compared to clinic population, indicating sampling bias that could affect model's real-world application.

2. Class imbalance handling presents methodological concern. Distribution between classes (44.5% born outside Canada vs 55.5% born Canada) isn't skewed, but researchers don't discuss techniques to address imbalance. Class weights or synthetic minority oversampling technique (SMOTE) could have improved model training. Lack of considerations may have influenced model's performance metrics, sensitivity, specificity, and could explain disparities in prediction accuracy between classes.

3. Cross-validation methodology shows limitations. Researchers used train-validation-test split (60:20:20) without implementing validation techniques like k-fold cross-validation or repeated sampling. This approach questions stability, reliability of reported performance metrics. Without multiple validation iterations, determining whether results remain consistent across data splits becomes difficult, raising possibility of random split artifacts affecting analysis.

4. Evaluation metrics, including sensitivity, specificity, PPV, NPV, need expansion. Missing confidence intervals for metrics hinders assessment of estimate precision. Researchers could have included F1 score or balanced accuracy, which suit class imbalance situations. Lack of ROC curves for regular expression approach limits performance comparison of methodologies across decision thresholds.

5. Feature engineering approach shows basic implementation, limiting model's pattern recognition capabilities. Researchers used bag-of-words approach for text features, missing opportunities for word embeddings implementation. EMR data temporal aspects, structured relationships between medical codes remained unused. Discussion omits feature selection methods, multicollinearity handling between features, raising questions about feature engineering process robustness. Enhanced feature engineering might have improved model performance.

**Do you want your identity to be public for this peer review?** For information about this choice, including consent withdrawal, please see our Privacy Policy..

Reviewer #1: No

Reviewer #2: No

**Figure resubmission:** While revising your submission, please upload your figure files to the Preflight Analysis and Conversion Engine (PACE) digital diagnostic tool, https://pacev2.apexcovantage.com/. PACE helps ensure that figures meet PLOS requirements. To use PACE, you must first register as a user. Registration is free. Then, login and navigate to the UPLOAD tab, where you will find detailed instructions on how to use the tool. If you encounter any issues or have any questions when using PACE, please email PLOS at figures@plos.org. Please note that Supporting Information files do not need this step. If there are other versions of figure files still present in your submission file inventory at resubmission, please replace them with the PACE-processed versions. **Reproducibility:** To enhance the reproducibility of your results, we recommend that authors of applicable studies deposit laboratory protocols in protocols.io, where a protocol can be assigned its own identifier (DOI) such that it can be cited independently in the future. Additionally, PLOS ONE offers an option to publish peer-reviewed clinical study protocols. Read more information on sharing protocols at https://plos.org/protocols?utm_medium=editorial-email&utm_source=authorletters&utm_campaign=protocols To enhance the reproducibility of your results, we recommend that authors of applicable studies deposit laboratory protocols in protocols.io, where a protocol can be assigned its own identifier (DOI) such that it can be cited independently in the future. Additionally, PLOS ONE offers an option to publish peer-reviewed clinical study protocols. Read more information on sharing protocols at https://plos.org/protocols?utm_medium=editorial-email&utm_source=authorletters&utm_campaign=protocols

---

## [Decision Letter · Decision Letter 1]

16 Mar 2026

Comparing regular expression and machine learning approaches to predict immigrant status from primary care electronic medical record data in Ontario, Canada

PDIG-D-24-00471R1

Dear Dr. Pinto,

We are pleased to inform you that your manuscript 'Comparing regular expression and machine learning approaches to predict immigrant status from primary care electronic medical record data in Ontario, Canada' has been provisionally accepted for publication in PLOS Digital Health.

Best regards,

Po-Chih Kuo, Ph. D.

Section Editor

PLOS Digital Health

**Additional Editor Comments (if provided):**

**Reviewer Comments (if any, and for reference):**

Reviewer's Responses to Questions

**Comments to the Author**

Reviewer #2: All comments have been addressed

publication criteria? Is the manuscript technically sound, and do the data support the conclusions? The manuscript must describe methodologically and ethically rigorous research with conclusions that are appropriately drawn based on the data presented.? Is the manuscript technically sound, and do the data support the conclusions? The manuscript must describe methodologically and ethically rigorous research with conclusions that are appropriately drawn based on the data presented.

Reviewer #2: Yes

3. Has the statistical analysis been performed appropriately and rigorously?

Reviewer #2: Yes

4. Have the authors made all data underlying the findings in their manuscript fully available (please refer to the Data Availability Statement at the start of the manuscript PDF file)?

The PLOS Data policy requires authors to make all data underlying the findings described in their manuscript fully available without restriction, with rare exception. The data should be provided as part of the manuscript or its supporting information, or deposited to a public repository. For example, in addition to summary statistics, the data points behind means, medians and variance measures should be available. If there are restrictions on publicly sharing data—e.g. participant privacy or use of data from a third party—those must be specified.requires authors to make all data underlying the findings described in their manuscript fully available without restriction, with rare exception. The data should be provided as part of the manuscript or its supporting information, or deposited to a public repository. For example, in addition to summary statistics, the data points behind means, medians and variance measures should be available. If there are restrictions on publicly sharing data—e.g. participant privacy or use of data from a third party—those must be specified.

Reviewer #2: Yes

5. Is the manuscript presented in an intelligible fashion and written in standard English?

Reviewer #2: Yes

Reviewer #2: The revised manuscript is well written, and I have no further comments. I recommend acceptance for publication.

**Do you want your identity to be public for this peer review?** For information about this choice, including consent withdrawal, please see our Privacy Policy..

Reviewer #2: **Yes:**Wisit CheungpasitpornWisit CheungpasitpornWisit CheungpasitpornWisit Cheungpasitporn
